# Perirectal Hematoma and Intra-Abdominal Bleeding after Stapled Hemorrhoidopexy and STARR—A Proposal for a Decision-Making Algorithm

**DOI:** 10.3390/medicina56060269

**Published:** 2020-05-29

**Authors:** Georgi Popivanov, Piergiorgio Fedeli, Roberto Cirocchi, Massimo Lancia, Domenico Mascagni, Michela Giustozzi, Ivan Teodosiev, Kirien Kjossev, Marina Konaktchieva

**Affiliations:** 1Department of Surgery, Military Medical Academy, 1606 Sofia, Bulgaria; ivanteodosiev@yahoo.com (I.T.); kirienkt@gmail.com (K.K.); 2Institute of Legal Medicine, University of Camerino, 62032 Camerino, Italy; piergiorgio.fedeli@unicam.it; 3Department of Surgical Science, University of Perugia, 06100 Perugia, Italy; roberto.cirocchi@unipg.it (R.C.); massimo.lancia@unipg.it (M.L.); 4Department of Surgical Science, Surgical Proctology Unit, Sapienza University of Rome, 00100 Rome, Italy; domenico.mascagni@uniroma1.it; 5Internal Vascular and Emergency Medicine and Stroke Unit, University of Perugia, 06100 Perugia, Italy; michela.giustozzi@unipg.it; 6Department of Gastroenterology, Military Medical Academy, 1606 Sofia, Bulgaria; marina.konaktchieva@yahoo.com

**Keywords:** stapled hemorrhoidopexy, stapled transanal rectal resection, perirectal hematoma

## Abstract

*Background and Objectives*: The present study aims to assess the effectiveness and current evidence of the treatment of perirectal bleeding after stapled haemorrhoidopexy. *Materials and methods*: A systematic literature review was performed that combined the published and the obtained original data after a search of PubMed, Web of Science, and SCOPUS. *Results*: The present systematic review includes 16 articles with 37 patients. Twelve papers report perirectal and six report intra-abdominal bleeding. Stapled hemorrhoidopexy (SH) was performed in 57% of cases (3 PPH 01 and 15 PPH 03), stapled transanal rectal resection (STARR) in 13%, and for 30% information was not available. The median age was 49 years (±11.43). The sign and symptoms of perirectal bleeding were abdominal pain (43%), pelvic discomfort without rectal bleeding (36%), urinary retention (14%), and external rectal bleeding (21%). The median time to bleeding was 1 day (±1.53 postoperative days), with median hemoglobin at diagnosis 8.8 ± 1.04 g/dL. Unstable hemodynamic was reported in 19%. Computed tomography scan (CT) was the first examination in 77%. Only two cases underwent the abdominal US, but subsequently, a CT scan was also conducted. Non-operative management was performed in 38% (n = 14) with selective arteriography and percutaneous angioembolization in two cases. A surgical treatment was performed in 23 cases—transabdominal surgery (3 colostomies, 1 Hartmann’ procedure, 1 low anterior resection of the rectum, 1 bilateral ligation of internal iliac artery and 1 ligation of vessels located at the rectal wall), transanal surgery (n = 13), a perineal incision in one, and CT-guided paracoccygeal drainage in one. *Conclusions*: Because of the rarity and lack of experience, no uniform tactic for the treatment of perirectal hematomas exists in the literature. We propose an algorithm similar to the approach in pelvic trauma, based on two main pillars—hemodynamic stability and the finding of contrast CT.

## 1. Introduction

After the initial presentation by Longo in 1998, the stapled haemorrhoidectomy (SH) has gained widespread popularity as a safe and effective surgical procedure for the treatment of grade III–IV hemorrhoids [1]. In this new technique, Longo suggested a circumferential rectal mucosectomy for a mucosal lifting [2].

The overall rate of complications is often underestimated [3], and bleeding is the second most common complication in patients with SH and is twice higher than in classic haemorrhoidectomy [4]. According to the most extensive series, the overall bleeding rate is 4.3%, and most frequently occurs immediately after surgery [5]. In half of them, it stops spontaneously, whereas only 0.43%, required re-intervention for surgical hemostasis under anesthesia [5]. This complication is associated with a more extended hospital stay, particularly in critically ill patients [6].

Postoperative bleeding can be intraluminal or perirectal [7]. In the significant number of cases, the postoperative bleeding is intraluminal from the submucosal vessels [8]. The reason is probably the large thickness of the visceral wall fold cut by the stapler, so the staples cannot adequately create adequate hemostasis [9]. Fortunately, this bleeding is effectively treated through an overstitching of the mechanical suture line at the time of the intraoperative check of the hemostasis [10].

The less frequent, but more troublesome complications, however, are perirectal, retroperitoneal, and intra-abdominal bleeding. Only a few case reports are published in the literature. In a recent systematic review of the literature, perirectal hematomas were not even included in the analysis of post-operative complications of SH [11]. Due to their rarity, no explicit treatment algorithm exists in everyday clinical practice and the current surgical guidelines. The present review is focused only on the cases with severe postoperative perirectal hematomas and intra-abdominal bleeding after SH and stapled transanal rectal resection (STARR) for obstructed defaecation. The aim is to determine the exact causes and to find out effective preventative measures and the most appropriate management plan. Last but not least, we propose useful advice for avoiding medico-legal severe consequences.

## 2. Materials and Methods

This review included only studies reported severe bleeding after SH or STARR—perirectal hematomas, retroperitoneal hematomas, or intra-abdominal bleeding, irrespective of the size, period of publication, and the language.

On 7 April 2020, we carried out a review of the literature on PubMed using the following searches:(“hemorrhage” [MeSH Terms] OR “hemorrhage” [All Fields] OR “bleeding” [All Fields]) AND stapled [All Fields] AND haemorrhoidopexy [All Fields];(“hemorrhage” [MeSH Terms] OR “hemorrhage” [All Fields] OR “bleeding” [All Fields]) AND stapled [All Fields] AND mucosectomy [All Fields];(“postoperative hemorrhage” [MeSH Terms] OR (“postoperative” [All Fields] AND “hemorrhage” [All Fields]) OR “postoperative hemorrhage” [All Fields] OR (“post” [All Fields] AND “operative” [All Fields] AND “hemorrhage” [All Fields]) OR “postoperative hemorrhage” [All Fields]) AND (“hemorrhoidectomy” [MeSH Terms] OR “hemorrhoidectomy” [All Fields])

In two other databases (SCOPUS and WOS), the searches were performed using the following keyword combinations:hemorrhage OR bleeding AND stapled OR haemorrhoidopexyhemorrhage stapled mucosectomyPostoperative hemorrhage hemorrhoidectomy

The Pubmed function “related articles” and Google Scholar database were used to search for further articles. Two authors (M.L. and F.P.) extracted the data independently, and any disagreements were resolved by a consensus meeting with a third review author (R.C.).

The Pubmed function “related articles” and Google Scholar database were used to search for further articles. Two authors (M.L. and F.P.) extracted the data independently, and any disagreements were resolved by a consensus meeting with a third review author (R.C.).

We developed four data grids compiling the characteristics of the patients included in the publications: excluded studies and reasons of exclusion, characteristics of the included studies (author, year of publication, country, type of study, number of patients enrolled, an indication to surgery, type of stapler), characteristics of patients (author, year of publication, the timing of bleeding, hemoglobin levels [mg/dL], signs and symptoms, hemodynamic instability), treatment (author, year of publication and type of treatment — Non-Operative Management, explorative laparotomy, drain of hematoma, angioembolization, diagnostic laparoscopy, colostomy, ligation of the arterial iliac artery, rectotomy).

Based on the data extracted from the included studies, we consider it impossible to conduct a meaningful meta-analysis of data. Consequently, we performed a narrative analysis of the outcomes of interest by summarizing retrieved data for each treatment.

## 3. Results

The search strategy identified 705 studies and 10 additional records identified through other sources. After de-duplication, 420 citations were screened, of which 397 were excluded based on title and abstract. For the remaining studies, the full texts were obtained and reviewed. Seven studies were excluded based on reasons listed in Table 1 [12,13,14,15,16,17,18]. Cumulatively, 17 articles (15 case reports and two case series) were considered relevant, and 38 patients were included (Figure 1, Table 2) [3,16,17,18,19,20,21,22,23,24,25,26,27,28,29,30,31,32,33].

Seven papers reported perirectal hematoma [3,17,18,20,21,22,23,24,25,26,27]. In seven articles, intraabdominal bleeding was described [28,29,30,31,32,33,34], two reported retroperitoneal hematoma [19,25]. In some cases, the first sign of perirectal bleeding was the intra-abdominal hemorrhage [28,29,31,32].

The most common procedure was the stapled hemorrhoidopexy (SH) in 58% (22 patients—4 PPH-01 and 15 PPH-03). A stapled transanal rectal resection (STARR) was performed in 13.5% (5 patients). In 29.7% (11 patients), the authors did not report the type of stapler (Table 2).

The characteristic of the patient is presented in Table 3.

The median age and the SD was 49 ± 11.43 years. The clinical manifestation was reported in all case reports (15 patients). The sign and symptoms were the abdominal pain (47%, 7/15), pelvic discomfort/pain (35.7%, 5/14), bleeding per rectum (21.5%, 3/14), urinary retention (14.2%, 2/14), back pain (7.1%, 1/14) and shock (7.1%, 1/14). One patient was asymptomatic (7.1%, 1/14).

The median time of bleeding was 1 ± 1.53 POD (postoperative days), and the median hemoglobin at diagnosis of bleeding was 8.8 ± 1.04 g/dL.

The data about hemodynamic status were reported in all case reports and the case series [4,5,6,7,8,9,10,11,12,13,14,15,16,17,18,19,20,21,22,23,24,25,26,27,28]: a hemodynamic instability was reported in a few cases (18.9%, 7/37) and a hemodynamic stabilization and resuscitation were needed.

The data about the radiological investigation was reported in 15 patients. The first examination was CT scan in 87% of cases (13/15), abdominal ultrasonography (US) was performed only in two cases (14.3%, 2/14). CT scan was conducted in two of the patients previously underwent the abdominal US. Only clinical examination was performed only in one case (7.1%).

The type of treatment was reported in all studies (Table 4). Nonoperative management was performed in 37% cases (14 patients of 38), with a selective angioembolization in two cases (5.4%).

The delayed drainage of perirectal hematoma was needed in 39% cases (15 patients) as follows: transanal rectoctomy in 13 cases (34%), perineal incisions in one case, and CT-guided paracoccygeal drainage in one.

An urgent laparotomic exploration was performed in nine cases as follows: three colostomies, one Hartmann’ procedure, one low anterior resection of the rectum, one suture of the rectum, one bilateral ligation of internal iliac artery, and one ligation of vessels located at the rectal wall.

## 4. Discussion

The possible cause for the perirectal bleeding is the full-thickness rectal wall resection extended to the perirectal adipose tissue (“pathological study of the resected rectal tissue may explain the evolution; the perirectal hematoma probably had its origin in the lesion of the blood vessels of the perirectal fat tissue that were partially transected by the stapling gun”.) [21]. Of note, SH is deemed to resect only redundant mucosa in contrast to the full-thickness resection in STARR. This can be a possible explanation of why there were only five with perirectal hematoma after the STARR procedure versus 32 after SH. However, a possible reason for this finding is the underreporting of the cases. CONTOUR^®^ TRANSTAR^™^ (Ethicon EndoSurgery, Cincinnati, OH, USA) was introduced into the practice to overcome the shortcomings of STARR by a tailored resection under visual control [35]. In 2008, a prospective European multicentre study reported bleeding requiring re-operation in 2/75 cases (3%), but the type of bleeding and intervention was not mentioned in detail [35]. In 2014, the European Registry reported only one case with perirectal hematoma required surgery in a series with 100 cases [36].

Other rare causes of late bleeding can be the pseudoaneurysms of the superior rectal artery [24,27,32]. Another, but not well studied and reported cause can be overlooked blood clotting disorders. Most frequently, the intra-abdominal bleeding is the consequence of peritoneal lacerations at the Douglass pouch [16,31,33] from the proximally expanding tension intramural rectal hematoma leading to disruption of the sigmoid wall with intra-abdominal bleeding [28]. The intraperitoneal lacerations may also be a result of the high placement of the stapler, which leads to full-thickness resection of the anterior rectal wall and sometimes peritoneal tears [31] or is a result of unrecognized deep enterocele, as in the case of Aumann et al. [34].

The perirectal hematoma after SH is first described in 2004 by Meyer et al. in a 52-year-old patient with grade III hemorrhoids [20]. On the first postoperative day, their patient manifests with symptoms of bleeding, and sudden anemia (8 g/L Hb, 24% Ht) without evident blood loss from the anus. The contrast abdominal CT revealed a giant perirectal hematoma (diameter 14 × 7 × 7 cm). The anoscopy showed no active bleeding. The patient was hemodynamically stable and underwent conservative medical treatment, as walls and bands well delimit the pelvic cavity, and the bleeding can be stopped by spontaneous tamponade (“Bei kreislaufstabiler Patientin fällt die Entscheidung zur primär konservativen Therapie mit dem Ziel: Sistieren der Blutung durch Eigentamponade”). This conservative treatment avoids the risk of making a temporary colostomy (“Damit auf ein temporäres Kolostoma verzichtet werden kann”).

The insidious nature of the rectal bleeding without an apparent external manifestation is most frequent when intra-abdominal bleeding occurs [28,31,32]. It should always be kept in mind because SH is often considered an “easy” and “routine” operation.

The perirectal hematomas are located in the perirectal space, within the mesorectum, in the “small” pelvis (“true pelvis”), and this extraperitoneal space is in communication with the mesosigma [28] and the retroperitoneal area [30]. Due to the rarity of the event, and the sparse literature on the topic, we suggest their management to follow the recent guideline for pelvic trauma [37]. Hemodynamic stability represents the primary factor in choosing the appropriate strategy. In this regard, Naldini describes two clinical scenarios [23]:progressive hematoma: the patient is hemodynamically unstable (i.e., in hemorrhagic shock), and it is necessary to carry out an emergency intervention to stop the bleeding [29]. This approach is part of the philosophy of Damage Control Surgery (DCS) with a subsequent definitive intervention after the stabilization of the hemodynamic status [37].stable pararectal hematoma: the patient is hemodynamically stable, so it is possible to perform a nonoperative treatment.

In unstable hemodynamic, the rectal packing with the placement of gauze strips into the rectal ampulla should be the initial step, followed by resuscitation and definitive hemostasis. The rectal packing differs from the pelvic one, where the strips are placed into the extraperitoneal space via a small incision of the abdominal wall and sometimes maybe a definitive treatment.

In hemodynamically stable or stabilizable patients, the contrast CT is the primary diagnostic modality, which can identify the bleeding source. In the case of active bleeding throughout the hematoma, the selective angiography of the lower mesenteric artery with super-selective embolization of the upper rectal artery is the method of choice [24,25,27,28]. The colonoscopy and rectoscopy are of only minimal value.

In stable hemodynamic, Non-Operative Management (NOM) can be successfully applied, even in intra-abdominal bleeding [32]. In the present review, 37% of the cases underwent NOM. The most worrying complication this approach is re-bleeding, which can occur in up to 34% [38]. It is a late and unpredictable event, often as a consequence of the development and rupture of a pseudoaneurysm of one of the branches of the upper rectal artery [24,27,32].

In the most extensive series in the literature describing staple line bleeding, surgical hemostasis was required in only 0.43% of the cases [5]. In contrast, the massive perirectal hematomas more frequently require surgery (63% of the presented cases). Almost half of them underwent transanal drainage of the hematoma to avoid the development of an infection and sepsis. Nevertheless, approximately 22% of the cases in the present review underwent transabdominal surgery, most frequently for an intramural tension hematoma devitalizing the rectal wall or causing disruption of the proximal bowel wall with intra-abdominal bleeding [28]. The most common procedure was the drainage of the hematoma and suture of the peritoneal laceration [34]. However, colostomy [24,30,33] and even rectal resection, with or without colostomy were required [29,32]. One case underwent bilateral ligation of the internal iliac arteries [23].

The laparoscopy is useful for both a diagnostic and therapeutic approach. It allows an assessment of the retroperitoneal hematoma, ruling out associated intra-abdominal injuries, washing out the abdominal cavity, draining the hematoma, and suturing of peritoneal lacerations if presented [16,19,20,25,26].

Based on the present review, we propose an algorithm to manage the perirectal hematomas and intra-abdominal bleeding after SH (Figure 2).

Apart from with the small sample size, another limitation of the present study is the possible underestimation of the actual rate of perirectal hematoma, probably because only the massive and clinically manifested case had been published. However, the present review was not intended to estimate the actual rate of perirectal hematoma but instead was focused only on the severe bleeding to describe the clinical characteristic and to propose an algorithm.

## 5. Conclusions

The perirectal hematomas and intra-abdominal bleeding are extremely rare but can be life-threatening. A possible cause for the perirectal bleeding is the full-thickness resection, which extends to the perirectal adipose tissue, a leak of the anastomosis, and pseudoaneurysms of the superior rectal artery. The intra-abdominal bleeding is due to peritoneal lacerations in the Douglass pouch or proximally expanding tension intramural rectal hematoma with disruption of the sigmoid wall. Another but not well studied and reported cause can be overlooked blood clotting disorders. 

An important characteristic is an insidious onset without an apparent external manifestation, which warrants close monitoring after SH. Because of the rarity and lack of experience, the management of the perirectal hematoma remains very complex, and no homogeneous treatment is described in the literature. According to the most extensive series in the literature, surgical hemostasis was required in only 0.43% of the cases with staple line bleeding. In contrast, the massive perirectal hematomas more frequently require surgery (63% of the presented cases). Almost half of them underwent transanal drainage of the hematoma to avoid the development of an infection and sepsis.

Herein we propose an algorithm to manage this complication, similar to the approach in pelvic trauma, based on two main pillars—the hemodynamic stability and the finding of contrast CT. In stable hemodynamic patients with the suspicion of perirectal hematoma, a contrast CT is needed with a NOM as the best choice. In contrast, in hemodynamically unstable patients, a DCS procedure is necessary.

## Figures and Tables

**Figure 1 medicina-56-00269-f001:**
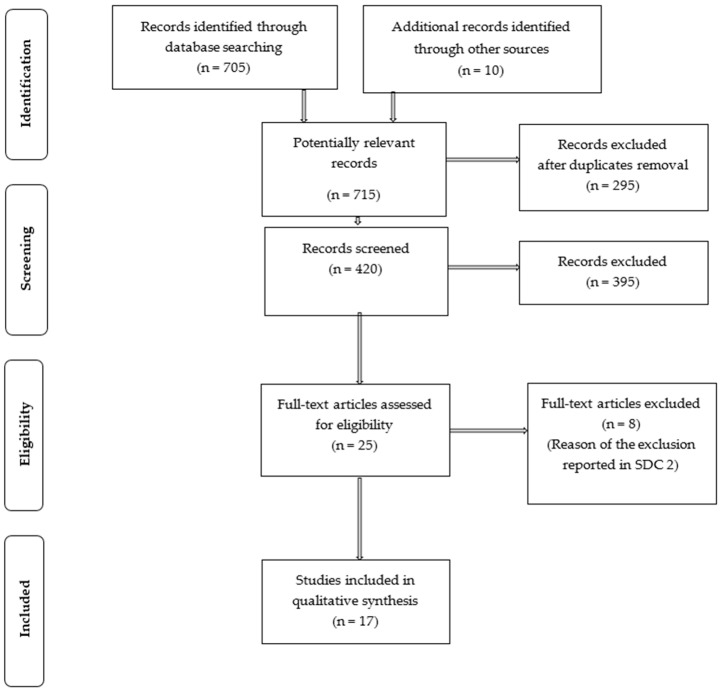
PRISMA Flow Diagram.

**Figure 2 medicina-56-00269-f002:**
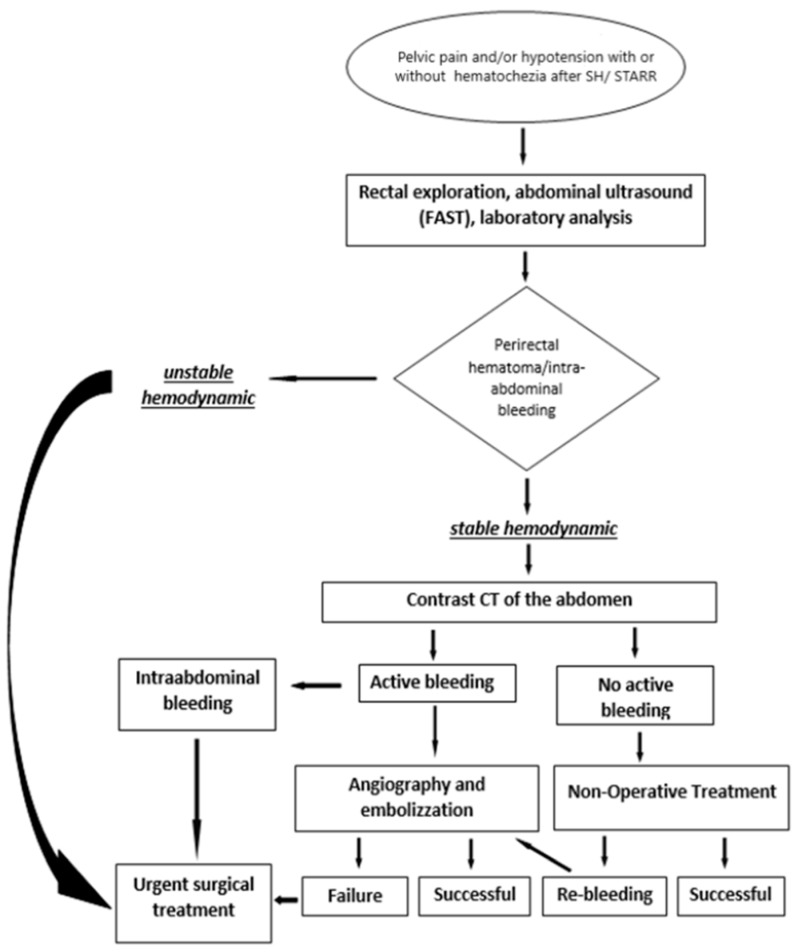
Algorithm for Management of perirectal hematomas and intra-abdominal bleeding after SH and STARR.

**Table 1 medicina-56-00269-t001:** The excluded studies and the reason for exclusion.

Author and Year of Publication	Reasons of Exclusion
Gallo 2020	A consensus statement on management and treatment of hemorrhoidal disease
Wang 2020	A comparative study between DST and PPH staplers in the treatment of grade III and IV hemorrhoids. None case of perirectal hematoma was reported
Lin 2019	A randomized clinical trial about partial stapled hemorrhoidopexy versus circumferential stapled hemorrhoidopexy for grade III to IV prolapsing hemorrhoids
Jeong 2017	A cohort study about partial Stapled Hemorrhoidopexy
Andreucetti 2014	A postoperative intra-abdominal bleeding after pile suturing
Lin 2012	A comparative study between partial stapled hemorrhoidopexy versus circular stapled hemorrhoidopexy for grade III–IV prolapsing hemorrhoids. None case of perirectal hematoma was reported
Arezzo 2011	A review of literature

DST—directional stapling technology, PPH—procedure for prolapse and hemorrhoids.

**Table 2 medicina-56-00269-t002:** Characteristic of the included studies.

Ref	Author and Year of Publication	Nation	Type of Study	Number of Patients Reported	Indication to Surgery	Type of Stapler (Number of Patients)
[30]	Ripamonti 2019	Italy	Case report	1	third-degree hemorrhoids	1 SH (PPH-03)
[27]	Rajkumar 2018	India	Case report	1	third-degree hemorrhoids	Other version of the gun
[26]	Ferrara 2018	Italy	Case report	1	third-degree hemorrhoids	1 SH (PPH-03)
[19]	Tebala 2016	Isle of Man	Case report	1	ODS	1 STARR
[25]	Cerullo 2015	Italy	Case report	1	ODS	1 STARR
[32]	Safadi 2014	Israel	Case report	1	third-degree hemorrhoids	1 SH
[24]	Shahzad 2013	Pakistan	Case report	1	third-degree hemorrhoids	1 SH
[33]	De Santis 2012	Italy	Case report	1	third-degree hemorrhoids	1 SH (PPH-03)
[23]	Naldini 2011	Italy	Multicentric cohort study	15	NR	4 SH (PPH-01) 9 SH (PPH-03) 2 STARR
[29]	Joyce 2012	Ireland	Case report	1	third-degree hemorrhoids	1 SH (PPH-03)
[22]	Chikkappa 2010	UK	Case report	1	second-degree hemorrhoids	1 STARR
[28]	Augustin 2009	Croatia	Case report	1	third-degree hemorrhoids	1 SH (PPH-03)
[31]	Blouhos 2007	Greece	Case report	1	third-degree hemorrhoids	1 SH (PPH-03)
[21]	Grau 2005	Spain	Case report	1	third-degree hemorrhoids	1 SH (PPH-01)
[3]	Oughriss 2005	France	Multicentric cohort study	8	third-degree hemorrhoids	NR
[20]	Meyer 2004	Germany	Case report	1	third-degree hemorrhoids	1 SH (PPH-01)
[34]	Aumann	Germany	Case report	1	third-degree hemorrhoids	1 SH (PPH-01)

SH—Stapled haemorrhoidopexy, PPH—procedure for prolapse and hemorrhoids, STARR—Stapled transanal rectal resection, ODS—obstructed defecation syndrome.

**Table 3 medicina-56-00269-t003:** Characteristics of the patients.

P	Author and Year of Publication	Timing of Bleeding (POD)	Hemoglobin Levels (mg/dL)	Sign and Symptoms	Hemodynamic Instability	CT Evaluation
[30]	Ripamonti 2019	1	7.6	Back and abdominal pain	No	Yes
[27]	Rajkumar 2018	3	NR	Bleeding per rectum	No	Yes
[26]	Ferrara 2018	<1	10	Abdominal and pelvic pain discomfort	No	Yes
[19]	Tebala 2016	2	8.8	Pelvic discomfort	No	Yes
[25]	Cerullo 2015	1	7.5	Asymptomatic	No	Yes
[32]	Safadi 2014	2	8.9	Abdominal pain	No	Yes
[24]	Shahzad 2013	4	NR	Bleed ing per rectum	No	Yes
[33]	De Santis 2012	6	NR	Bleed ing per rectum	No	Yes
[23]	Naldini 2011	NR	NR	NR	No 10 Yes 5	NR
[29]	Joyce 2012	<1	8.8	Shock	Yes	No
[22]	Chikkappa 2010	1	9.3	Urinary retention	No	No
[28]	Augustin 2009	1	NR	Abdominal and pelvic pain	No	Yes
[31]	Blouhos 2007	1	6.2	Abdominal and pelvic pain	Yes	Yes
[21]	Grau 2005	<1	7.8	Urinary retention—pelvic pain	Yes	Yes
[3]	Oughriss 2005	NR	NR	NR	No	NR
[20]	Meyer 2004	1	8.9	Abdominal pain	No	Yes
[34]	Aumann 2004	1	NR	Abdominal pain	No	Yes

POD—postoperative day, CT—computed tomography, NR—not reported

**Table 4 medicina-56-00269-t004:** Treatment.

Ref	Author and Year of Publication	Type of Treatment (Number of Patients)	Surgical Treatment/Interventional Radiological Treatment (Number of Patients)
[30]	Ripamonti 2019	NOM	-
[27]	Rajkumar 2018	NOM	Angioembolization
[26]	Ferrara 2018	NOM	Angioembolization
[19]	Tebala 2016	NOM	Diagnostic laparoscopy
[25]	Cerullo 2015	NOM	Diagnostic laparoscopy
[32]	Safadi 2014	NOM	-
[24]	Shahzad 2013	NOM	Angioembolization
[33]	De Santis 2012	Explorative laparotomy	Colostomy
[23] [29] [22] [28] [31] [21] [3] [20]	Naldini 2011	Explorative laparotomy (4)	Colostomy (2)
			Bilateral ligation of the internal iliac arteries
			Ligation of vessel on the rectal wall
		NOM (5)	Diagnostic laparoscopy
			Angioembolization
		Drain the haematoma (6)	Rectotomy (4)
			Perineal incisions
			CT-guided paracoccygeal
[34]	Joyce 2012	Explorative laparotomy	Colostomy
[30]	Chikkappa 2010	Drain the haematoma	Rectotomy
[27]	Augustin 2009	Explorative laparotomy	Hartmann’s resection
[26]	Blouhos 2007	Explorative laparotomy	Low Anterior Resection of the Rectum
[19]	Grau 2005	NOM	-
[25]	Oughriss 2005	Drain the haematoma (8)	Rectotomy (8)
[32]	Meyer 2004	NOM	-
[24]	Aumann 2004	Explorative laparotomy	Suture of the rectum

NOM—Non-Operative Management.

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
