# Peer review of "Perirectal Hematoma and Intra-Abdominal Bleeding after Stapled Hemorrhoidopexy and STARR—A Proposal for a Decision-Making Algorithm"

_medicina, 2020, doi:10.3390/medicina56060269_

Round 1
Reviewer 1 Report
The manuscript of Popivanov et al. deals with the question of the strategies in the treatment of severe intra-abdominal or perirectal bleeding after stapled hemorrhoidopexy. This question is clinically relevant and an overview of high value is to be developed.
Formally
The manuscript is appropriate in length and writing style. However, the following problems arise from the methological access to the questions. The authors refer to this work as a systemic review. According to Cook, these strategies include a comprehensive search of all potentially relevant articles and the use of explicit, reproducible criteria in the selection of articles for review (Cook, Ann Int Med 1997). The defined processing of the extraction criteria of the found literature is missing. Should the authors insist on describing their work as a systemic review, a paragraph must be included in the Methods section, as the found literature has been extracted and processed in a defined way.
The search for literature is incomplete, for example the publications of Aumann and colleagues from 2004 (Aumann Tech Coloproctol 2004) are missing.
Content
The summary work deals with the important question of treatment after intraabdominal or perirectal bleeding after stapled hemorrhoidopexy.
First, there is speculation about the causes of a severe bleeding complication after hemorrhoidopexy. This section completely ignores the fact that the most common cause of bleeding will be an overlooked blood clotting disorder. Faulty surgical procedures are probably much rarer compared to undetected blood clotting disorders. Another section speculates on the surgical technical causes. Here it is disregarded that intra-abdominal bleeding is not necessarily caused by an excessively high positioning of staples, but can also be the result of a very deep enterozele, especially in women.
An algorithm is presented to treat the complications. This is obviously the most important aspect of this review and the authors should be encouraged to improve this algorithm. If there is clinical evidence of a serious complication after stapling surgery, the first question is whether the patient is hemodynamically stable. If there is hemodynamic stability, the diagnosis should be carried out by means of abdomen CT, it is important to point out in the overview work that endoluminal diagnostics by colonoscopy and rectoscopy are of only very very limited value. In hemodynamically unstable patients, first the restoration of the circulation and then the fastest possible diagnosis by emergency CT is mandatory. Interventional procedures may help to treat bleeding. It is very much to discuss which surgical interventions are clearly useful at this time.
In summary, the authors should be encouraged to improve overview work according to the above-mentioned notes.
Reviewer 2 Report
The submitted work deals with a rare but severe and difficult to treat complication of transanal anorectal resection by the use of a stapler. And for that the Authors must be congratulated.
However several comments could be made:
Major remarks
- To mix a review and a case-report is unusual and not recommended generally. As the latter does not bring a lot I would delete it to focus solely on the review.
- A number of studies notably those producing the largest number of cases, come from the previous decade. A word on "learning curve" of these procedures would be welcome.
- Legal issues are a different topic and should not be treated here as it covers all complications occurring with staplers.
- A reminder that PPH is not deemed to resect full rectal wall would be most welcome and then a comment on the fact that PPH led to more cases than STARR procedure would be helpful (different number of procedures done? different pathophysiology? different reporting?). A word on TranStar another rectal stapling procedure and consequences with regard to the current topic has also to be considered.
Point by point remarks
- Abstract: STARR is not mentioned in the Aims
- Methods: 7 April 2020. Too short a time for such an analysis. A better definition of "perirectal haematoma" should be given as probably these here described are the only ones that gave signs (underestimation of the occurrence following the discussed procedures)
- Intraabdominal bleeding is not (sufficiently) reported from studies
- Rectal packing needs more description and when discussing Damage Control to make a distinction with pelvic packing required in major pelvic trauma.
- Laparoscopic approach: used in this setting? or is this only a hypothesis?
- Conclusion: the first sentence is misleading. The perirectal haematoma rate is not 4.3%. If it would have been the procedure would have been probably stopped
Author Response
Please see the attachment.

This manuscript is a resubmission of an earlier submission. The following is a list of the peer review reports and author responses from that submission.